# Multisite phosphorylation drives phenotypic variation in (p)ppGpp synthetase-dependent antibiotic tolerance

Elizabeth A. Libby [1,2,3], Shlomi Reuveni[2,4,5,6] & Jonathan Dworkin[1]*

Isogenic populations of cells exhibit phenotypic variability that has specific physiological consequences. Individual bacteria within a population can differ in antibiotic tolerance, but whether this variability can be regulated or is generally an unavoidable consequence of stochastic fluctuations is unclear. Here we report that a gene encoding a bacterial (p)ppGpp synthetase in *Bacillus subtilis*, *sasA*, exhibits high levels of extrinsic noise in expression. We find that *sasA* is regulated by multisite phosphorylation of the transcription factor WalR, mediated by a Ser/Thr kinase-phosphatase pair PrkC/PrpC, and a Histidine kinase WalK of a two-component system. This regulatory intersection is crucial for controlling the appearance of outliers; rare cells with unusually high levels of *sasA* expression, having increased antibiotic tolerance. We create a predictive model demonstrating that the probability of a given cell surviving antibiotic treatment increases with *sasA* expression. Therefore, multisite phosphorylation can be used to strongly regulate variability in antibiotic tolerance.

[1] Department of Microbiology and Immunology, College of Physicians and Surgeons, Columbia University, New York, NY 10032, USA. [2] Department of Systems Biology, Harvard Medical School, Boston, MA 02115, USA. [3] Wyss Institute for Biologically Inspired Engineering, Harvard University, Boston, MA 02115, USA. [4] School of Chemistry, Tel-Aviv University, 6997801 Tel-Aviv, Israel. [5] Center for the Physics and Chemistry of Living Systems, Tel Aviv University, 6997801 Tel Aviv, Israel. [6] The Sackler Center for Computational Molecular and Materials Science, Tel Aviv University, 6997801 Tel Aviv, Israel. *email: jonathan.dworkin@columbia.edu

Many bacterial phenotypes, including antibiotic tolerance and virulence, often reflect the phenotype of a subset of the population rather than the average behavior[1,2]. Subpopulations of bacteria can arise through purely stochastic processes, as well as by regulatory and signaling pathways[3]. Theoretically, one way to create phenotypic diversity via a signaling pathway is multisite phosphorylation, in which each successive phosphorylation changes the activity of a protein[4,5]. However, it has not been experimentally shown in bacterial populations that multisite phosphorylation regulates variation in gene expression between cells, and subsequently, the emergence of phenotypic diversity. Recently, multisite phosphorylation of transcription factors have been observed in pathways involved in antibiotic tolerance and virulence[6], suggesting that dynamics of multisite phosphorylation could have particular physiological relevance.

Bacterial signaling is often characterized in the context of two-component signal transduction systems (TCS) that generally consist of a histidine kinase that phosphorylates a response regulator on a single residue, which then acts as a transcription factor[7]. The stimulus-dependent response of this type of signaling system architecture has been analyzed theoretically[8,9] and experimentally[10,11], with little cell-to-cell variability observed (as quantified by CV), regardless of inducer level. This suggests that extensive cell-to-cell variability is not a general feature of bacterial TCS. However, some notable exceptions have been found for two-component systems with more complex architectures, such as the broad distribution of gene expression in the E. coli TorS/TorR regulon[12] which has recently been shown to be an important factor for cell survival during oxygen depletion[13]. The network architecture of bacterial signal transduction systems may therefore play an underappreciated role in the dynamics and survival of bacterial populations.

In addition to TCS, bacteria also have eukaryotic-like (also called Hank's type) Ser/Thr kinase – phosphatase pairs with homology to eukaryotic systems that perform reversible phosphorylation on Ser and Thr residues[14]. One particular subfamily of these systems appears to be universally conserved across Gram-positive bacteria and plays key roles in growth and virulence for many clinically important pathogens including the streptococci, S. aureus, M. tuberculosis, E. faecalis, and others[6,15]. Genetic and proteomic studies indicate that these Ser/Thr kinases can perform transcriptional regulation of key cellular processes involved in antibiotic tolerance and persistence through multisite phosphorylation of transcription factors. However, to date, the consequences of multisite phosphorylation for gene regulation at the single-cell-level has not been quantified. In this context the model gram-positive bacterium B. subtilis presents a comparatively straightforward system to quantify the contribution of the additional Ser/Thr phosphorylation in vivo: the homologous kinase-phosphatase pair is PrkC/PrpC, and it has been verified to regulate gene expression through additional phosphorylation of a response regulator.

It has been apparent for over 60 years that bacterial populations contain rare cells that display increased phenotypic resistance to antibiotics[16]. These cells, presumed to be quiescent, have been implicated in antibiotic treatment failure in genetically susceptible bacterial infections[17]. To date, it remains unclear to what extent the appearance of these rare cells is subject to regulation. Emerging evidence strongly implicates elevated levels of the nucleotide second messenger (p)ppGpp as a causative agent of quiescence in many bacterial species[18,19]. (p)ppGpp downregulates essential cellular processes such as transcription, translation, and DNA replication[20]. Although the precise mechanism of (p)ppGpp synthesis and its direct cellular targets vary between bacterial species, highly elevated levels of (p)ppGpp confer a quiescent state to the bacterial cell. As many antibiotics target active cellular processes, the resulting quiescent cells exhibit increased antibiotic tolerance[21], suggesting that cell-to-cell variability in (p)ppGpp may be involved in phenotypic resistance to antibiotics.

The mechanistic origin of cell-to-cell variability in (p)ppGpp levels across bacterial populations remains a major open question. To date, this has been best studied in E. coli, in the context of the RelA (p)ppGpp synthetase and the SpoT hydrolase[22]. In contrast, other bacterial species often possess dedicated (p)ppGpp synthetases, termed small alarmone synthetases (SAS), in addition to bi-functional synthetase-hydrolases[23]. These SAS proteins can be activated transcriptionally[20], suggesting that cell-to-cell variability in (p)ppGpp levels could originate in the transcriptional regulation of the synthetases themselves. In the Gram-positive bacterium B. subtilis, three distinct proteins, RelA, SasA, and SasB synthesize (p)ppGpp[24]. B. subtilis RelA is a bi-functional (p)ppGpp synthetase-hydrolase, and both SasA and SasB are dedicated synthetases. Although relA and sasB transcripts are both readily detectable during log phase growth, sasA (formerly ywaC) transcripts are found at considerably lower levels. However, sasA is inducible by certain classes of cell-wall-active antibiotics[25,26], and its induction by alkaline shock increases the cellular levels of ppGpp[24]. Since sasA expression stops growth[27], SasA-mediated (p)ppGpp synthesis provides a mechanism to induce cellular quiescence in response to environmental stresses. To date, SasA is only known to be regulated transcriptionally, so significant cell-to-cell variability in sasA expression could produce physiologically relevant cell-to-cell variability in (p)ppGpp levels. The pre-existing distribution of sasA expression may therefore be critical in predicting the relative survival of cells under conditions that do not specifically induce sasA.

In this work, we demonstrate that sasA expression displays physiologically relevant amounts of extrinsic noise, although the average level of sasA expression is very low during growth under non-inducing conditions. Furthermore, we find that both the distribution of sasA expression and the frequency of outliers are strongly regulated by the activity of a highly conserved eukaryotic-like Ser/Thr kinase system and its subsequent multisite phosphorylation of a transcription factor. Using quantitative analysis of the full distributions of sasA expression, we find that multisite phosphorylation is responsible for exponentially regulating the abundance of cells with a given level of SasA, and we generate a predictive model for sasA-expression-dependent antibiotic tolerance.

## Results

**sasA expression exhibits high levels of extrinsic noise.** While the population average level of sasA expression during growth is extremely low[25], the average behavior may mask important phenotypic variation between cells. We therefore generated a transcriptional reporter for sasA ($P_{sasA}$-yfp) to study the population at the single-cell level. Surprisingly, there was considerable cell-to-cell variability in $P_{sasA}$-yfp (coefficient of variation, CV ~ 4.95 ± 0.42, mean ± SEM), with most cells having very low expression, and rare cells showing significantly higher levels of expression (Fig. 1a). Quantification of YFP fluorescence revealed that a small fraction of the population had much higher (>~10×) levels of fluorescence than the mean, and rare cells had ~100×. Note that a typical bacterial gene has a CV in the range 0.1–1[28–30]. Consistently, we measured the B. subtilis gene veg to have a CV of 0.4 ± 0.02 (mean ± SEM, Supplementary Fig. 1).

The high levels of cell-to-cell variability in sasA expression could be caused by intrinsic noise from the promoter itself, or by extrinsic noise originating in an upstream process[31]. To differentiate between these mechanisms, we used a strain with dual fluorescent reporters, $P_{sasA}$-yfp and $P_{sasA}$-mcherry (Fig. 1b).

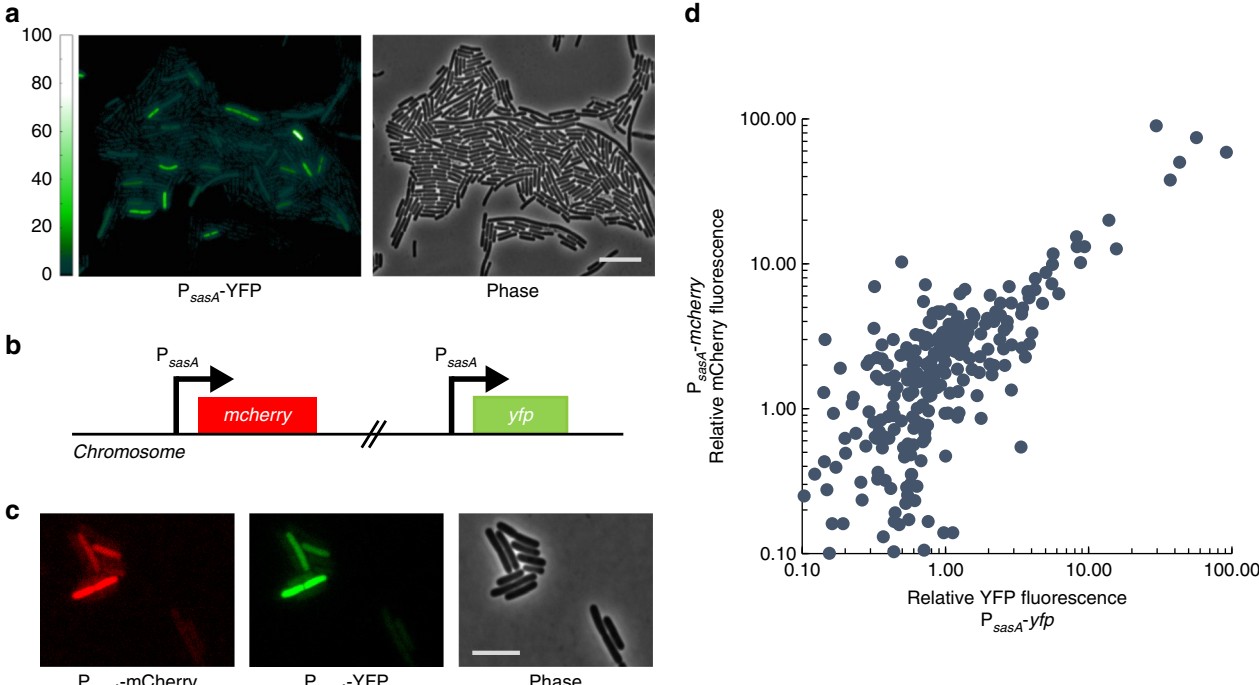

**Fig. 1** *sasA* exhibits cell-to-cell variability in expression. **a** P$_{sasA}$-*yfp* expression in log phase culture. Left**:** YFP image. Color scale indicates the fold change relative to the population average. Right**:** Phase contrast. Scale bar indicates 10 μm. **b** Schematic of the dual-color transcriptional reporter strain used to characterize the noise in *sasA* expression. Two copies of the *sasA* promoter driving different reporters (P$_{sasA}$-*yfp*, P$_{sasA}$-*mcherry*) are inserted at separate ectopic loci. **c** Evidence for extrinsic noise in *sasA* expression. Images of single cells from the dual reporter strain with significant fluorescence shown in mCherry (Left), YFP (Center), and phase contrast (Right) channels. Scale bar indicates 5 μm. **d** Quantification of cellular fluorescence intensities for single cells in the experiment described in **b**, **c**. Values were normalized relative to the population mean value for each channel. Data shown was measured on ~580 cells with detectable fluorescence in both channels. The fluorescence of the two reporters have a Pearson's correlation coefficient of $r$ ~ 0.90 ± 0.08 (mean ± SEM, 4 experiments). For each reporter, the CV ~ 4.95 ± 0.42 (mean ± SEM, 4 experiments)

Expression of the dual reporters in individual cells was highly correlated (Pearson's correlation coefficient, $r$ ~ 0.90 ± 0.08, mean ± SEM; Spearman's correlation coefficient ~ 0.78 ± 0.06, mean ± SEM), demonstrating that the noise was largely extrinsic to the promoter (Fig. 1c, d).

Some variability in P$_{sasA}$ is expected to originate in natural variability in protein expression and accumulation between cells in a population[32]. Therefore, we compared the variability observed in P$_{sasA}$-*yfp* to a presumably unrelated promoter known to be constitutively active during log phase growth, P$_{veg}$-*mcherry*[33] (Supplementary Fig. 1). We found that YFP and mCherry levels were not highly correlated, suggesting that the high levels of variability in *sasA* expression are largely caused by a *sasA*-specific pathway.

We then tested a previously characterized *sasA* regulator, the sigma factor σ$^M$ (SigM)[34], that is required for *sasA* expression[26] (Supplementary Fig. 2A). *sigM* expression levels were generally detectable at low levels across the population (Supplementary Fig. 2B). We also found that *sasA* expression did not correlate strongly with *sigM* expression (Supplementary Fig. 2C, D) demonstrating that SigM levels alone do not predict variability in *sasA* expression. To our surprise, we also found that the relatively rare outliers in *sigM* expression were not predictive of the outliers we observed in *sasA*. Since the outliers in *sasA* expression would likely exhibit the strongest *sasA*-dependent physiological effects on the cell, we sought to determine the regulatory factors responsible for regulating their frequency in the population.

**The Ser/Thr kinase PrkC represses *sasA* expression through WalR.** Another potential regulator of *sasA* is the WalR transcription factor observed to bind the *sasA* promoter in a genome-wide screen[35]. WalR is the response regulator of the essential WalRK two-component system and is activated by phosphorylation of Asp-53 by WalK[36]. Once phosphorylated, WalR can activate and/or repress genes in its regulon. A reversible second phosphorylation on WalR Thr-101 by the eukaryotic-like Ser/Thr kinase-phosphatase pair PrkC/PrpC[37] further increases WalR activity at both activating and repressing sites[38]. In rich media (LB), the multisite phosphorylation of WalR affects gene expression (e.g., enhanced activation of *yocH*) specifically in post-log phase[38]. However, in the commonly used defined minimal media S7-glucose, there is a consistent PrkC-dependent effect on the population average level of *yocH* expression throughout log phase (Supplementary Fig. 3).

*sasA* is known to be activated by antibiotics such as bacitracin[25] through σ$^M$ activation. We first tested whether the PrkC/PrpC–WalR system regulates *sasA* at the population level to determine if WalR activates or represses *sasA*. We found that PrkC activity represses *sasA* expression through WalR Thr101~P during bacitracin treatment (Supplementary Fig. 4). Based on these bulk measurements, we developed a model for *sasA* regulation (Fig. 2a) in which PrkC activity further potentiates WalR-repressing activity at *sasA* through a second phosphorylation of WalR at Thr-101. However, it remained unclear to what extent multisite phosphorylation of WalR affects pre-existing cell-to-cell variability and physiologically relevant outliers in *sasA* expression under non-inducing conditions.

**PrkC regulates noise in *sasA* through WalR Thr-101 phosphorylation.** Cell-to-cell variability in gene expression can be tuned by changing repressor-binding affinities[39,40], suggesting that multisite phosphorylation of WalR may play a critical role in setting the

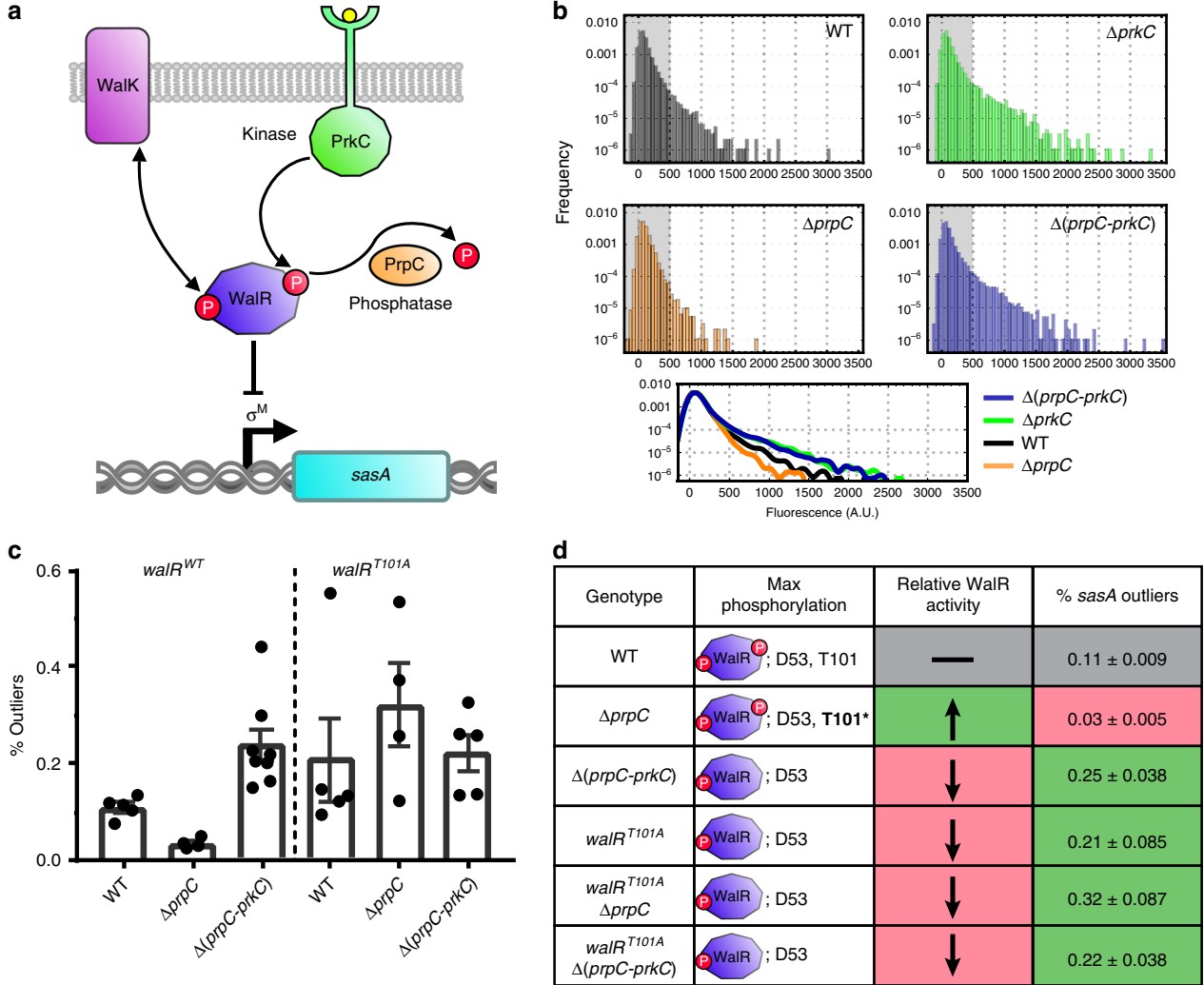

**Fig. 2** The Ser/Thr kinase PrkC and phosphatase PrpC regulate cell-to-cell variability in *sasA*. **a** Model for PrkC-dependent regulation of *sasA*. WalR binding to the *sasA* promoter represses *sasA* expression. WalR activity is primarily controlled by phosphorylation on Asp-53 by its cognate histidine kinase WalK, and secondarily by phosphorylation on Thr-101 by the Ser/Thr kinase PrkC. WalR Thr-101~P can be dephosphorylated by the phosphatase PrpC. Thr-101-P further enhances the repressing activity of WalR Asp-53-P, resulting in lower expression of *sasA* under conditions with high Thr-101~P (e.g., Δ*prpC*). Under conditions lacking Thr-101~P (e.g., Δ*prkC*), the increased repression of *sasA* is relieved. **b** Histograms demonstrating PrkC-dependent cell-to-cell variability in *sasA*. Top: P$_{sasA}$-*yfp* reporter activity was quantified by flow cytometry in wild type (WT, gray), Δ*prpC* (orange), Δ*prkC* (green), and Δ(*prpC-prkC*) (blue) backgrounds. Shaded region of each plot indicates the range of cellular autofluorescence observed. Each histogram was computed from data on ~$3.0 \times 10^4$ events. Bottom: Smoothed, overlaid histograms of the data shown above for comparison. **c** Percentage of outliers in each genetic background. At least 4 independent experiments, similar to and including the representative one shown in **b**, were performed in *walR*$^{WT}$ (Left) and *walR*$^{T101A}$ (Right) backgrounds. Each experiment was normalized to a control and outliers were defined as cells above a fixed threshold level of normalized fluorescence (~1250 A.U.). Dots represent the percentage of each population that is above the threshold; bars and lines represent the mean and SEM, respectively. **d** Summary of each genotype, its effect on the maximal occupancy of the two known WalR phosphosites, and the expected effect on WalR activity relative to WT. For each genotype the mean percentage of outliers in *sasA* expression (shown in **c**) is also summarized. Note that in the Δ*prpC* background, the Thr-101 phosphorylation is stabilized (denoted as **T101***)

observed distribution of *sasA* expression across the population. To test this, we measured the distribution of *sasA* expression in wild type (WT) cells and compared it to genetic backgrounds that alter the phosphorylation state of WalR: Δ*prpC* (no phosphatase, high levels of T101~P), and Δ*prkC* (no kinase, no detectable T101~P) (Fig. 2b, Supplementary Fig. 5). Qualitatively, in the Δ*prpC* background, the frequency of cells with high levels of *sasA* expression was strongly reduced, whereas it was strongly increased in the Δ*prkC* background. The PrpC-dependent effect on *sasA* expression requires PrkC, since the distribution of *sasA* expression in a strain lacking both the kinase and phosphatase (Δ(*prpC-prkC*)) is very similar to a strain lacking just the kinase.

We first sought to quantify the effect of WalR multisite phosphorylation on the frequency of "outliers": cells with a level of *sasA* expression above a fixed threshold in each population. We therefore compared independent measurements of the distribution of *sasA* expression in WT, Δ*prpC*, and Δ(*prpC-prkC*) backgrounds (Fig. 2c, left) and found that PrkC significantly affects the mean frequency of outliers >8 fold by this measure (*walR*$^{WT}$, Δ*prpC* vs. Δ(*prpC-prkC*): **p*-value ~ 0.004, Kolmogorov-Smirnov test). We repeated the measurements in a *walR*$^{T101A}$ background (Fig. 2c, right) and found that PrkC no longer has a significant effect on the mean frequency of outliers in the phosphosite mutant background (*walR*$^{T101A}$, Δ*prpC* vs. Δ(*prpC-prkC*): *p*-value ~ 0.56,

ns, Kolmogorov-Smirnov test). These results are consistent with increased WalR activity by Thr-101 phosphorylation causing increased repression of *sasA*, and thereby regulating the frequency of *sasA* outliers (Fig. 2d). Furthermore, heterologous expression of PrkC was sufficient to reduce the frequency of outliers observed in the Δ*prkC* background (Supplementary Fig. 6A). Heterologous expression of PrkC was also able to further reduce the variability to below that observed in the Δ*prpC* background, approaching the level of cellular autofluorescence (Supplementary Fig. 6B). This suggests that at least some of the remaining variability in the Δ*prpC* background arises due to incomplete saturation of WalR T101~P.

This definition of outliers, however, relies on the definition of a cutoff threshold and therefore does not fully address how multisite phosphorylation affects the distribution of *sasA* expression across the entire population. To quantify the effect of PrkC on the distribution of cell-to-cell variability in *sasA* expression, we deconvolved the measured data from the cellular autofluorescence (see Methods section). This resulted in autofluorescence-free distributions of *sasA* expression, allowing better quantitative comparison of expression between genetic backgrounds (Fig. 3a). This deconvolution method uses only the first two moments (i.e., the mean and variance) of the observed distributions of fluorescence. As such, the auto-fluorescence free distributions are relatively insensitive to the observed "outliers" in each distribution, but makes a statistical prediction for those frequencies. To verify the accuracy of the predictions, these calculated distributions were re-convolved with the cellular

autofluorescence and the reconstructed data set compared to the original data (Supplementary Fig. 7). Calculation of the relative enrichment of cells with a given level of *sasA* in each genetic background revealed that maximal T101~P (Δ*prpC*) results in exponential changes in the relative abundance of cells at a given level of *sasA* compared to the absence of T101~P (Δ*prkC*) (Fig. 3b).

Together, the model and the outlier analysis in Fig. 2 suggest that PrkC-dependent regulation of the distribution of *sasA* expression requires the second WalR phosphosite at Thr-101. To test this, we repeated the deconvolution procedure for a strain expressing a WalR mutant that lacks the Thr-101 phosphosite, WalR T101A, and found that the PrkC-dependent effect on *sasA* expression is indeed WalR Thr-101-dependent (Fig. 3c, Supplementary Fig. 8A). Thus, multisite phosphorylation is responsible for the exponential depletion of cells with medium to high levels of *sasA* expression in the Δ*prpC* background (Fig. 3b). We note that exponential depletion has a strong effect on the frequency of outliers with high levels of *sasA* expression, and only modestly effects the first two moments of the distribution (mean and variance). Consistently, we found that although the effect on the frequency of outliers is large, the CV of the distributions measured by flow cytometry changes by only ~30% (For the data shown in Fig. 3b: WT ~ 2.87, Δ*prpC* ~ 2.2, Δ*prkC* ~ 2.61, Δ(*prpC-prkC*) ~ 2.65).

We then measured how intermediate levels of multisite phosphorylation regulate the distribution of *sasA* expression

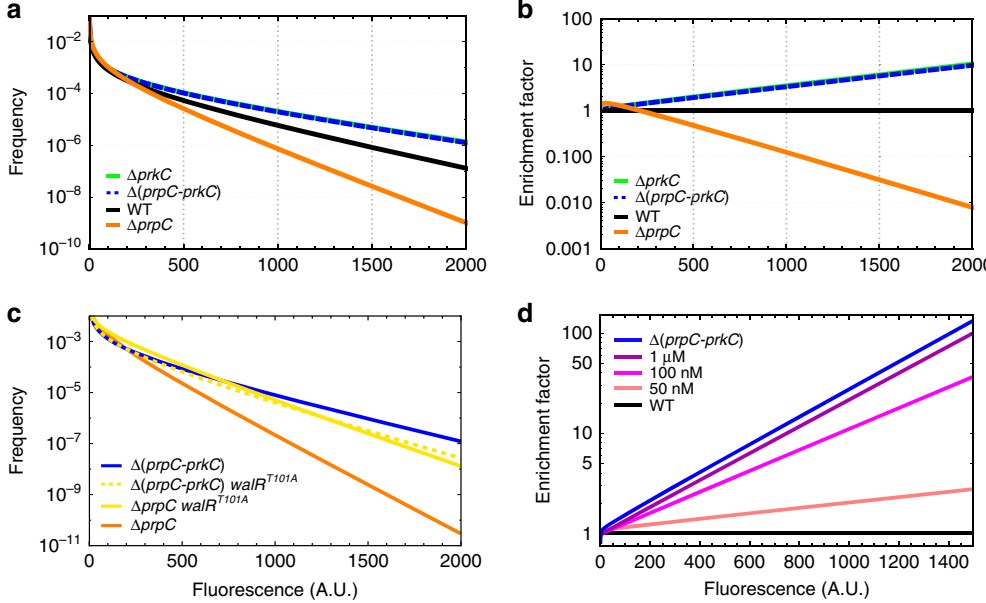

**Fig. 3** Multisite phosphorylation causes exponential changes in the abundance of cells with a given level of *sasA* expression. **a** Functional fits of autofluorescence-free distributions of *sasA* expression. A deconvolution algorithm was used to remove the contribution of autofluorescence from the measured distributions of *sasA* expression shown in Fig. 2b. Validation of the fits are shown in Supplementary Fig. 7. **b** Effect of PrkC on the distribution of *sasA* expression. For each genotype in **a** the frequency of cells with a given level of *sasA* expression was normalized against the equivalent frequency in the WT population. On this semi-logarithmic plot, straight lines indicate that the enrichment factor changes ~exponentially with fluorescence. **c** PrkC-dependent regulation of cell-to-cell variability in *sasA* is WalR Thr-101-dependent. P_{sasA}-*yfp* reporter activity was quantified by flow cytometry in a Δ*prpC* walR^{T101A} (yellow, solid) background and compared to Δ*prpC* (orange), Δ(*prpC-prkC*) (blue), and Δ(*prpC-prkC*) walR^{T101A} (yellow, dashed) backgrounds in the same experiment. Plots were generated by applying the deconvolution method described in **a** to the measured data in Supplementary Fig. 8A. **d** Gradual inhibition of PrkC results in progressive enrichment of cells with elevated levels of *sasA*. P_{sasA}-*yfp* reporter activity was quantified by flow cytometry during treatment with increasing concentrations of staurosporine: 0 (solvent only; black), 50 nM, 100 nM, and 1 μM (shades of magenta), in otherwise WT populations. For reference, a Δ(*prpC-prkC*) (blue) population treated with solvent only is shown; note that the effect of staurosporine saturates at this level. Plots were generated by applying the deconvolution method described in **a** to the measured data in Supplementary Fig. 8B, C. For each condition the frequency of cells with a given level of *sasA* expression was normalized against the equivalent frequency in the WT population. As in **b**, straight lines indicate that enrichment increases ~exponentially with fluorescence. Enrichment becomes more pronounced as the concentration of staurosporine increases and saturates at the level of complete inhibition ~Δ(*prpC-prkC*)

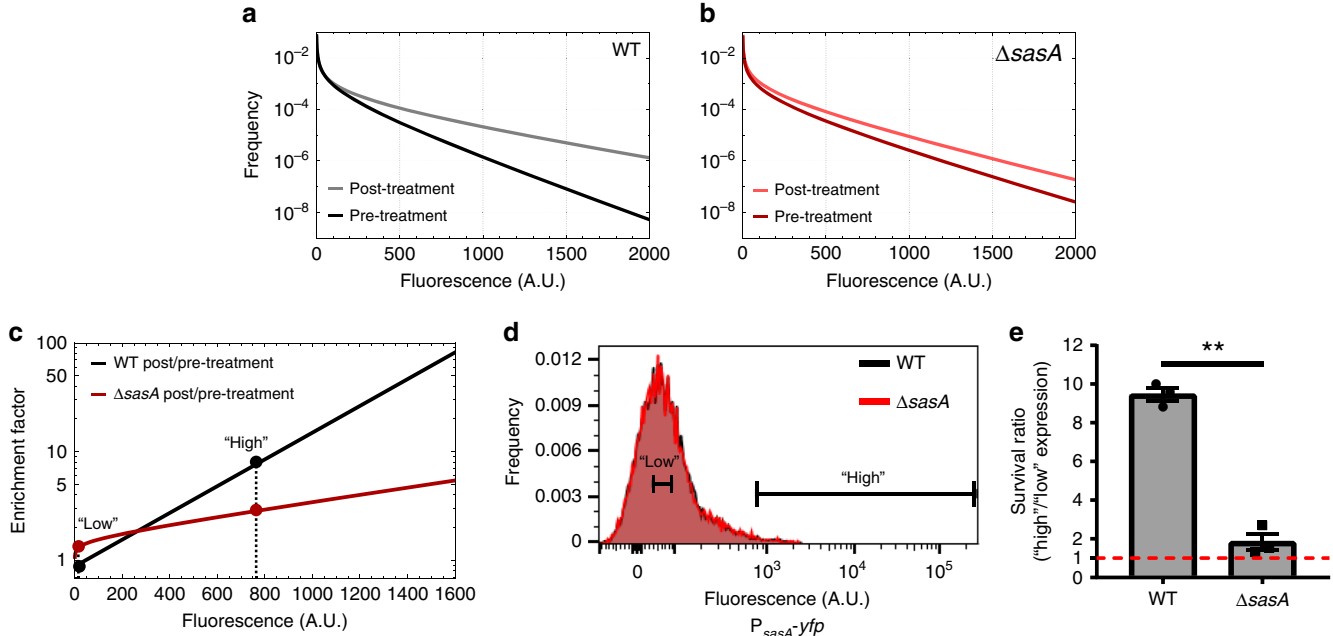

**Fig. 4** Cells with elevated *sasA* expression preferentially survive antibiotic treatment. **a** Autofluorescence-free distributions of *sasA* expression before (dark gray) and after (light gray) ciprofloxacin treatment in an otherwise WT background based on the data shown in Supplementary Fig. 10A. **b** Autofluorescence-free distributions of *sasA* expression before (dark red) and after (light red) ciprofloxacin treatment in a Δ*sasA* background based on the data shown in Supplementary Fig. 10B. **c** Enrichment plot following antibiotic treatment for WT and Δ*sasA* populations. For each genotype in **a**, **b** the frequency of cells with a given level of P*sasA*-*yfp* after antibiotic treatment was normalized against the corresponding frequency prior to antibiotic treatment. Dashed lines and dots indicate the average values used in **d**, **e**. **d** Histograms of P*sasA*-*yfp* fluorescence in WT (black) and Δ*sasA* (red) backgrounds measured by flow cytometry prior to sorting into "low" (4.1 ± 1.8) or "high" (779.5 ± 134.9) expression groups (relative mean ± SEM, 3 experiments). Each histogram is comprised of data obtained from ~3.0*$10^4$ events and is from a representative experiment used in **e**. **e** Relative survival of the "high" and "low" expression populations in **d** after treatment with ciprofloxacin for ~3.5 h as in **a**, **b**. Bars indicate the mean survival ratio of the "high" to "low" expressing populations, lines indicate the SEM. The data shown represents 3 independent experiments (paired *t*-test, *p*-value ~ 0.009, **). The red dashed line indicates a survival ratio of 1, indicating no advantage

using the kinase inhibitor staurosporine to progressively inhibit PrkC activity[41] (Supplementary Fig. 8B; 9). The distributions of *sasA* expression were again deconvolved (Supplementary Fig. 8C), and we calculated the relative enrichment of cells with a given level of *sasA* fluorescence at increasing concentrations of staurosporine (Fig. 3d). Titration of PrkC activity resulted in exponential enrichment of cells with a given level of *sasA*. Therefore, even small changes in PrkC activity result in large changes in the abundance of "outliers", cells with unusually high levels of *sasA*.

**sasA expression level continuously predicts antibiotic tolerance.** Cell-to-cell variability in (p)ppGpp production has been proposed to result in cell-to-cell variability in antibiotic survival[42–45]. However, a direct and quantitative relationship between the expression of a transcriptionally regulated (p)ppGpp synthetase and the probability of survival for an individual cell has not been demonstrated. We therefore sought to determine if cells with pre-existing high levels of *sasA* preferentially survive antibiotic exposure, and if so, to provide a model for how the level of *sasA* expression influences the probability of survival for a given cell.

We used ciprofloxacin, a DNA gyrase inhibitor that does not significantly increase the population average level of *sasA* expression[25]. We measured (Supplementary Fig. 10) and deconvolved (Fig. 4a, b) the distributions of *sasA* expression (P*sasA*-*yfp*) both pre-ciprofloxacin and post-ciprofloxacin treatment that results in ~99% killing in both WT and Δ*sasA* backgrounds when measured at the bulk population level (Supplementary Fig. 11). We note that, importantly, the starting

distributions of P*sasA*-*yfp* are very similar in both genetic backgrounds, allowing a direct comparison. Using these distributions, we calculated the relative enrichment of cells with a given level of *sasA* following antibiotic treatment (Fig. 4c), yielding a simple model for the effect of antibiotic treatment on the distribution of *sasA* expression (see Methods section). Because survival after antibiotic treatment can be affected by many processes, we separated out the size of the *sasA*-dependent effect by using a Δ*sasA* mutant as a control. WT populations exhibited a significant increase in the fraction of cells with elevated levels of *sasA* (Fig. 4c). This effect is strongly reduced in the Δ*sasA* background, demonstrating that *sasA* has a significant contribution to survival after ciprofloxacin treatment. We repeated this analysis on Δ(*prpC-prkC*) and Δ(*prpC-prkC*) Δ*sasA* strains to determine if a similar effect holds for the phosphorylation mutant that generates an increased frequency of cells with high levels of *sasA* expression (Supplementary Fig. 12). We found that Δ(*prpC-prkC*) populations have a ~50% increase in cells with high levels of *sasA* both pre-ciprofloxacin and post ciprofloxacin treatment. Consistent with the results on WT populations, the effect is partially *sasA*-mediated. This suggests that rare, pre-existing cells with increased probability of survival are more abundant in Δ(*prpC-prkC*) than in WT populations.

We then sought to determine whether our enrichment model reflects the probability of survival for cells as a function of *sasA* expression. Since SasA is a (p)ppGpp synthetase, the *sasA*-dependent enrichment we observe post-ciprofloxacin treatment could be due to an expression-dependent probability of surviving antibiotic treatment. Alternately, since the measured increase in mean fluorescence was relatively small (~2 fold), ciprofloxacin

could act in a complex, expression-dependent, manner to generate the observed post-treatment distribution of *sasA* expression without affecting survival. To differentiate between these hypotheses, we used FACS to sort bacteria prior to ciprofloxacin treatment from both WT and Δ*sasA* populations into "high" (upper ~1%) and "low" (~average) P*sasA*-*yfp* expression groups (Fig. 4d). Importantly, sorting cells by *sasA* expression level prior to ciprofloxacin treatment allows more sensitive measurements of the survival advantage conferred by SasA than traditional bulk population level CFU measurements. In addition, cells were measured and sorted prior to antibiotic treatment and assayed by CFUs to avoid confounding survival measurements with changes in cell physiology, fluorescent protein expression, or lysis in response to ciprofloxacin. Following ciprofloxacin treatment (as in Fig. 4a, b, Supplementary Fig. 10), the relative survival of "high" and "low" expression cells, or the survival ratio, was assayed by plating for CFUs (Fig. 4e). We observed that cells with high levels of *sasA* expression had a survival advantage of 9.5 ± 0.6 for WT, but only 1.8 ± 0.7 in Δ*sasA* background, demonstrating that cells with high levels of *sasA* expression prior to antibiotic treatment preferentially survive.

The average fluorescence cutoff values used in the FACS experiments, low: 4.1 ± 1.8 and high: 779.5 ± 134.9 (mean ± SEM, 3 experiments), were then used as inputs for a model where the enrichment of cells with increased levels of *sasA* (Fig. 4c) is caused by increased survival (see Methods section). The model yielded good agreement with the results of the FACS experiments: it predicted relative survival ratios of ~9 for wild-type, and ~2 for Δ*sasA*, respectively, compared to the measured values of ~9.5 ± 0.6 (WT) and 1.8 ± 0.7 (Δ*sasA*) (Fig. 4e). We found that relaxing the experimental cutoff for the "high" threshold to ~60% of the value in Fig. 4d resulted in a strong reduction of the survival advantage for WT: 2.3 ± 0.7 fold relative survival for WT, and 0.9 ± 0.4 fold for Δ*sasA* (mean ± range, 2 experiments) (Supplementary Fig. 13). This is in reasonably good agreement with a model prediction of 3.7 and 2.0-fold, respectively. Therefore, the enrichment of cells with elevated levels of *sasA* post-ciprofloxacin treatment can be largely attributed to the increased survival probability of pre-existing cells in the population with elevated *sasA* expression.

Taken together, our results demonstrate that an important consequence of PrkC-dependent multisite phosphorylation of WalR is the regulation of cell-to-cell variability, or noise, in the WalR regulon gene *sasA*. By comparing the full distributions of gene expression, we demonstrate that this effect is not just confined to the regulation of outliers in gene expression above an arbitrary threshold, but has an exponential effect on the relative abundance of cells with a given level of expression in the population. By analyzing the full distributions of expression, we are also able to demonstrate that *sasA* expression continuously affects the antibiotic tolerance of individual cells: specifically, the survival probability during a fixed course antibiotic treatment. This model (see Methods section) is consistent with cell sorting experiments that explicitly demonstrate that the observed distributions are a consequence of differential survival probabilities.

## Discussion

Antibiotic tolerance is believed to be an important factor in the failure of antibiotic treatments and a key step toward the development of antibiotic resistance[46]. Noise in expression of genes that regulate cellular quiescence are hypothesized to play an important role in cell-to-cell variability in tolerance. We therefore sought to trace the origin of the cell-to-cell variability in

expression of the (p)ppGpp synthetase *sasA* and determine if it can be regulated by genetic or chemical means. Noise in gene expression can be conceptually separated into intrinsic and extrinsic noise. Although it is difficult to design strategies to specifically target events generated by intrinsic noise, extrinsic noise may have upstream regulatory pathways that can be modulated. Therefore, it is significant that the cell-to-cell variability in *sasA* was dominated by extrinsic noise at high levels of expression (Fig. 1) that have the strongest effect on antibiotic tolerance (Fig. 4). Furthermore, since multisite phosphorylation is responsible for setting the observed distribution of cell-to-cell variability (Figs. 2, 3), this regulatory pathway could be a novel antibiotic target.

Multisite phosphorylation can expand the range of a protein's function, generating both switch-like[47,48], and graded[49,50] changes in average activity. In contrast, here we observed only minimal changes in the average levels of *sasA* expression as a function of PrkC activity, but measured up to a ~100-fold effect on the frequency of "outliers", cells with particularly high levels of expression (Fig. 3b). This response was shown to be graded, rather than switch-like, likely arising as a consequence of the integration of signals from two distinct signaling systems. A single phosphorylation at WalR Asp-53 strongly, but imperfectly, represses the *sasA* promoter. The addition of the second phosphorylation at Thr-101 by a distinct signaling system then acts as a second input to further regulate WalR. Interestingly, even small changes in activity of the second system result in marked changes in the frequency of outliers. Heterologous expression of PrkC is capable of reducing the variability observed to nearly cellular autofluorescence, but does not eliminate it completely. This remaining variability in *sasA* may be due to PrkC overexpression still being unable to completely saturate WalR phosphorylation, intrinsic noise at the promoter, or as yet unidentified sources. This demonstrates that PrkC activity is likely heterogeneous in our culture condition, but does not pinpoint the cause or origin of the variability. For example, from this study we cannot rule out that cells with high levels of *sasA* expression originate in a PrkC-dependent manner from the colony itself.

Transcriptional regulation of outliers in eukaryotes has been shown to be predictive of which cancer cells survive drug treatment[51]. Here, we found that transcriptional regulation by multisite phosphorylation is also critical for setting the pre-existing distribution of survival probabilities for cells within a bacterial population. Distinct from bacterial persistence, which is characterized by bi-phasic killing, these survival probabilities reflect antibiotic tolerance or the killing kinetics during a relatively short, fixed time-course, antibiotic treatment[52]. In the Δ*sasA* background, we observed a much weaker dependence of antibiotic tolerance on *sasA* expression. This residual dependence is consistent with previous results that have implicated many global processes in antibiotic tolerance including heterogeneity in growth state[53–55] and enhanced expression of drug efflux pumps[56–58]. This is also consistent with the relatively weak correlation in expression between *sasA* and the constitutive promoter *veg* (Supplementary Fig. 2). Indeed, it remains to be seen precisely how cellular physiology changes in a *sasA*-expression dependent manner. SasA has been shown to be important for ribosome dimerization in *B. subtilis*[27] and for survival during envelope stress in *S. aureus*[59]. More generally, various cellular processes are known to be directly and indirectly affected by rising (p)ppGpp levels including inhibition of DNA primase activity[60], and reduction in intracellular GTP pools[61] thereby downregulating rRNA transcription[62]. As our results show that antibiotic survival increases continuously with *sasA* expression, they suggest that SasA exerts a continuous effect proportional to its level on physiological processes that effect ciprofloxacin killing. Therefore,

multisite phosphorylation may provide a "bet-hedging" strategy to regulate the phenotypic diversity of a bacterial population, serving as a broadly useful mechanism to tune the frequency of rare phenotypes that facilitate survival under adverse conditions.

## Methods

**Strain construction**. For a listing of strains used in figures, see Supplementary Table 1. All strains are derivatives of *B. subtilis* 168 *trpC2* unless otherwise noted. For additional details of strain and plasmid construction, see Supplementary Data 1 and 2, respectively. For a table of oligos used in this study, see Supplementary Table 2.

**Media and culture conditions**. *B. subtilis* cultures were grown in the chemically defined minimal medium S7[60], modified from[63], supplemented with trace elements[64] and L-tryptophan to early log phase. In each experiment, strains were streaked out from frozen stocks on LB Lennox and grown overnight (~15 h) at 37 °C. Single colonies were used to inoculate 3 ml liquid cultures in S7 (supplemented with inducers as indicated) and cultures were grown to early log phase (OD$_{600}$ ~ 0.1–0.2). Dilutions for OD$_{600}$ matching, if required, were no more than 1:3.

**Microscopy and image analysis**. Microscopy was performed on live cells immobilized on 1% agarose pads prepared with S7 media. Imaging was performed using a Nikon 90i or a TE2000 microscope with a Phase contrast objective (CFI Plan Apo Lambda DM ×100 Oil, NA 1.45), an X-Cite light source, a Hamamatsu Orca ER-AG, and the following filter cubes: YFP (ET Sputter 500/20×, Dm515, 535/30 m), and mCherry (ET Sputter Ex560/40 Dm585 Em630/75). To generate representative fields of log phase cultures, cultures were concentrated ~20–100× immediately prior to imaging. Images were processed using Fiji[65].

Quantitative measurement of gene expression was performed similarly to ref. [66]. Briefly, phase contrast and fluorescence images (e.g., YFP, mCherry) were acquired of well separated bacterial cells immobilized on agarose pads. The resulting image stacks were segmented based on the phase contrast image, and the corresponding average florescence per pixel within each cell was calculated for each fluorescence channel using Matlab. A non-fluorescent control strain, treated with antibiotics as needed, was used as a control to subtract background and autofluorescence in each channel.

**Luminescence assays**. Luminescence assays were performed similarly to as described[38]. Briefly, the cultures were initially grown to early log phase in a roller drum at 37 °C. 150 μl of each culture was loaded into 96 well plates and 100 μg/ml bacitracin added as indicated. Measurements were performed in a Tecan Infinite 200 plate reader. Luminescence and OD$_{600}$ were measured at 5 min intervals with continuous shaking and the values of all samples at a defined time point after the luminescence reached ~steady state, about 1 h post-treatment, are reported.

**Flow cytometry and cell sorting**. Cultures were grown to early log phase and diluted 2–4 fold with additional S7 media to obtain the optimal density for flow cytometry. The resulting samples were vortexed vigorously prior to measurement to disrupt aggregates. Flow cytometry and sorting were performed on a BD Biosciences FACS Aria II-SORP or a Miltenyi MACSquant VYB (Supplementary Figures 6 and 12 only). YFP was detected using a blue laser (488 nm) with a 525/50 dichroic, and a 505 long pass filter for both flow cytometers. mCherry was detected using a yellow/green laser (561 nm) with a 582/15 dichroic, and a 570 long pass filter (BD Aria only). Fluorescence values were quantitatively compared between experiments by rescaling each experiment by the mean fluorescence of a control sample. Sorting was performed with a 70 nm nozzle at 70 PSI. Detection voltages were set such that the non-fluorescent control had a median value of ~100. Unless otherwise noted, flow cytometry data shown is representative of at least 3 biological replicates.

The sorting thresholds were chosen such that the "high" expression gate corresponds to approximately 1% of the starting population and can be sorted in ~10 min. Higher thresholds than this result in very long sort times due to decreases in overall sorting efficiency, and also appeared to introduce unwanted variability in cell physiology. Lower thresholds (e.g., as in Supplementary Fig. 13), resulted in a strong reduction of the size of the effect.

**Survival assays**. ~1.5 × 10$^4$ cells were sorted by fluorescence and dispensed into equal volumes of chemically defined growth media (S7). Cultures were incubated in a roller drum at 37 °C for 10 min, then treated with 500 ng/ml ciprofloxacin for 3.5 h. Serial dilutions of each population were plated for colony forming units (CFUs) on LB and grown overnight at 30 °C and survival ratios were calculated by comparing the CFU values between the high and low fluorescence groups. In each experiment, wild-type and mutant strains were tested in parallel.

**Autofluorescence deconvolution method and validation**. For each flow cytometry experiment, data from both *sasA* reporter strains and a nonfluorescent control were measured. As nonfluorescent controls for each genotype were verified to have similar autofluorescence, typically a single control measurement was performed for each experiment. Raw measurements of fluorescence intensities from transcriptional reporter strains have two contributions: (i) the transcriptional reporter for *sasA* expression (e.g., P$_{sasA}$-*yfp*), the signal of interest; and (ii) background fluorescence originating from other sources. Measuring the statistics of the background fluorescence using a non-fluorescent control allowed extraction (deconvolution) of the *sasA* transcriptional reporter signal from the total raw measurement consisting of the sum of this signal and background fluorescence[67].

We note that on our data sets, brute force deconvolution in absence of any prior knowledge of the signal statistics is quite noisy and error prone. However, in bacteria, the statistics of protein copy numbers is known to be well described by the Gamma distribution, which is a great advantage as this distribution is fully specified in terms of its mean and variance. Given the measured mean and variance of the raw fluorescence intensity and of the autofluorescence, the mean and variance of the signal can be estimated by assuming that the signal from the *sasA* transcriptional reporter is not correlated with background fluorescence. (This assumption is supported by fluorescence microscopy experiments showing that cell size was not well correlated with *sasA* expression, and by computationally verifying that simple forms of dependency are inconsistent with the observed data.) The mean and variance of fluorescence from the *sasA* transcriptional reporter could then be uniquely determined, as described in more detail in the mathematical details of the deconvolution procedure.

As mentioned above, we used a Gamma distribution to model the distribution of the fluorescence signal for a given mean and variance (we separately verified that other distribution choices, such as lognormal, did not capture the data as well). The Gamma distribution is routinely applied to fit gene expression data,[30] but here we additionally verified that it closely mimics the underlying signal coming from the reporter by an in-silico re-convolution of the Gamma-distribution-fitted autofluorescence-free signal with measured background fluorescence. This was done by drawing two random fluorescence intensities: one from the estimated Gamma distribution for the clean (autofluorescence-free) signal and one from the measured autofluorescence distribution. Adding these two random contributions gave a single, re-convolved, data point. Repeating this procedure for a number of times equaling the size of the measured data set yielded the full re-convolved data set. The empirical distributions of the measured and re-convolved data sets were then compared (Supplementary Fig. 7) and found to be almost indistinguishable from those that were originally measured (Supplementary Fig. 3). No smoothing was applied.

**Mathematical details of deconvolution procedure**. The raw fluorescence intensity ($R$) measured from transcriptional reporter strains were assumed to have two statistically independent contributions. One coming from the signal of interest ($S$), and one coming from other sources, collectively treated here as noise ($N$). One then has

$$R = S + N \tag{1}$$

for every measurement taken, and averaging over all measurements immediately gives the following relation

$$<R> = <S> + <N> \tag{2}$$

between the mean fluorescence values of the raw fluorescence intensity, the signal of interest, and the noise. Statistical independence then asserts that the variance in the fluorescence intensity could also be decomposed in a similar way

$$Var(R) = Var(S) + Var(N). \tag{3}$$

These relations, and the fact $<R>$, $<N>$, $Var(R)$, $Var(N)$ could all be directly computed from measured data allowed us to estimate the mean, $<S>$, and variance, $Var(S)$, of the signal of interest. The full distribution of the signal of interest was then assumed to be well described by a Gamma distribution, since this has previously been shown to correctly describe gene expression data in bacteria[30] and we have moreover verified that this assumption is internally consistent with the data (see Autofluorescence deconvolution method and validation). The probability density function describing $S$ was therefore modeled as

$$f(s) = \frac{1}{\Gamma(k)\theta^k} s^{k-1} e^{-s/\theta}, \tag{4}$$

where $\Gamma(\cdot)$ denotes the gamma function, and the parameters $k$ and $\theta$ are uniquely set by their relations to the mean

$$<S> = k\theta, \tag{5}$$

and variance

$$Var(S) = k\theta^2, \tag{6}$$

of the signal.

**Mathematical details of enrichment model**. Deconvolved fluorescence intensities of cells before, $f_b(s)$, and after, $f_a(s)$, ciprofloxacin treatment were modeled by the Gamma distribution to test the hypothesis that they are related via

$$f_a(s) = C \cdot f_b(s) \cdot p(s), \tag{7}$$

where $p(s)$ stands for the probability of a cell with fluorescence intensity $s$ to survive the prescribed treatment and $C$ is a normalization constant assuring that

$$\int_0^\infty f_a(s)ds = \int_0^\infty C \cdot f_b(s) \cdot p(s)ds = 1. \tag{8}$$

Following treatment, we define the enrichment factor (in the probability density) of cells with fluorescence level $s$ as

$$EF(s) = f_a(s)/f_b(s). \tag{9}$$

One could then write the ratio between the survival probability of cells with fluorescence $s_1$ and $s_2$ (Survival ratio) in the following way

$$\text{Survival ratio} = \frac{p(s_1)}{p(s_2)} = \frac{C \cdot p(s_1)}{C \cdot p(s_2)} = \frac{f_a(s_1)/f_b(s_1)}{f_a(s_2)/f_b(s_2)} \tag{10}$$

This relation was then used to predict survival ratios and compare with data as described in the main text.

**Significance testing for % outlier data shown in Fig. 2**. In Fig. 2c at least 4 independent experiments measuring *sasA* expression were performed in WT, $\Delta prpC$, and $\Delta(prpC\text{-}prkC)$ in otherwise $walR^{WT}$ and $walR^{T101A}$ backgrounds. For each genotype, the % of cells above a fixed fluorescence threshold was measured and is shown as black dots. To test if the % values obtained for each genotype are statistically significant, we treated the independent measurements of %s as members of a distribution and performed Kolmogorov-Smirnov testing to determine the likelihood that the % outliers observed for a comparison genotype pair could be drawn from the same distribution.

**Reporting summary**. Further information on research design is available in the Nature Research Reporting Summary linked to this article.

## Data availability
The data that support the findings of this study are provided in the paper itself and associated source data files.

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

## Acknowledgements

We thank Eric Brown for the strain EB1385, Isaac Plant (Silver lab) for pIP384 and IP563, and Amir Figueroa at the Microbiology and Immunology Core Facility at Columbia University for assistance with flow cytometry and cell sorting. This work was supported by NIH GM114213 and a BWF Investigators in the Pathogenesis of Infectious Disease award to J.D., a grant from the Department of Systems Biology at Harvard Medical School to E.A.L., and S.R. was supported by NIH GM095784 and gratefully acknowledges support from the Azrieli Foundation.

## Author contributions

Conceived and designed the experiments E.A.L. and J.D. E.A.L. performed the experiments. E.A.L. and S.R. analyzed data. Contributed reagents/materials/analysis tools: E.A.L. and S.R. Wrote the paper: E.A.L., S.R., and J.D.

## Competing interests

The authors declare no competing interests.
