## [Peer Review File · Nature Communications]

Reviewers' comments:

Reviewer #1 (Remarks to the Author):

This paper explores cell-to-cell variability of *sasA*, which encodes a ppGpp synthase in *B. subtilis*. The authors show that *sasA* transcription in single cells has a very broad distribution across a population of cells and results from high extrinsic noise. They further show that the noise is primarily due to second-site phosphorylation at T101 in the response regulator WalR, with T101 phosphorylation controlled by the kinase PrpK and the phosphatase PrpC. Finally, they provide evidence that the noise has important physiological consequences, namely increased antibiotic tolerance. Over all I found the paper to be well written, the arguments compelling, and the results very interesting. The work provides a striking example of drug tolerance emerging from infrequent transcriptional events, a subject that is still in its nascent stages in both prokaryotic and eukaryotic biology. In addition, the regulation of cell-to-cell variability by second site phosphorylation of a response regulator seems quite novel.

I have only a few comments.

1) Fig. 2A. Why does the arrow that comes off of WalR split in two with one arrow pointing to PrkC? The figure makes it look like PrkC plays a role in dephosphorylating WalR, but there is no mention of this in the text. Some explanation or clarification in the figure caption should be provided.

2) Fig. 2B The histograms showing the distribution of autofluorescence should be shown (perhaps put it in the supplement). I realize that CDF for the autofluorescence is shown in Fig. S5 but it would be helpful to see the actual data binned as in Fig. 2B so that the reader can get a rough sense of the contribution of autofluorescence to histograms. Was the autofluorescence measured for each of the four strains shown in Fig. 2B or was a single strain used and the autofluorescence was assumed to be the same for the other strains?

3) Lines 174-175 I don't understand how the Komogorov-Smirnov test was used to compute p-values for the data in Fig. 2C. I understand how this test is used to compare distributions, but in this case, the data consists of a mean frequency of outliers based on a fixed threshold. Based on the data, the claim that PrkC has a significant effect on the mean frequency of outliers looks reasonable to me, but I suggest adding some sentences to the Materials and Methods explaining how the p-value was computed using the K-S test.

4) To "deconvolve" the autofluorescence from the signal, why didn't the authors do an actual deconvolution using the measured autofluorescence distribution, instead of only removing the mean and standard deviation components?

5) I do not understand how the autofluorescence-free distributions were "re-convolved" with the autofluorescence data to generate Fig. S6. Was the autofluorescence-free distribution convolved with the actual measured distribution of autofluorescence and then smoothed? If not, where did the distribution for the autofluorescence data come from to take the convolution? Some more details on this should be included.

Reviewer #2 (Remarks to the Author):

The manuscript presents observations of the high level of extrinsic noise in the expression of *sasA*, a ppGpp synthetase and the analysis of its origin and consequences. The authors dissect the role played by other factors known to affect *sasA*. They first measure the levels of SigM and conclude that it does not account for the variability. Then they analyze the role of the phosphorylation of

WalR by the two component system PrpC/PrpK. They show that the deletion of prpC reduces the number of highly expressing cells. In order to compare not only the outliers but the whole distribution, they deconvolve each distribution with the strain's auto-fluorescence signal. They expose the wt and sas deletion strain to an antibiotic and show a difference in the ratios of survival of high and low expressing yfp which suggest that the highly expressing sasA cells may be more tolerant than the low expressing sasA cells.

Whereas the role of phosphorylation for generating variability is an interesting idea, here the analysis of mutants merely shows that the few outliers are missing but does not explain the interesting observation of the very large CV of sasA. The impact of such variability on persistence is potentially very interesting. However, there is no direct demonstration of the role of sasA or its phosphorylation on tolerance as detailed in the comments below.

1. The effect of the phosphorylation mutants seems to be measurable only for a very small portion of the population. What are the CVs of the distributions for the mutants?
2. The survival of the wt and delta sasA are not mentioned, only the ratios of high to low, which seem arbitrary. How do the ratios change if the thresholds are changed?
3. Importantly, in order to confer tolerance, the survival itself needs to be mentioned, not only the ratios. Is the sas deletion mutant less tolerant? Are any of the other mutants tested more or less tolerant than the wt? This information is crucial in order to understand the role of sasA noise in tolerance as well as the role played by phosphorylation.
4. The authors conclude that the increased sasA expression in the FACs is due to the enrichment of pre-induced cells. However, the FACS does not enrich for surviving bacteria, does it? Ciprofloxacin killing typically does not result in cell lysis so that both dead and viable cells will be measured by FACS.
5. Fig. S1: it is not clear why the data is presented on a linear vs log scale, and not in log-log scales as in Fig. 1. Also, the correlation coefficients should be calculated over the whole range and compared to the correlation of the two sas promoters (0.9 Pearson). BTW, because of the outliers, it may be better to calculate the Spearman coefficients.
6. Fig. S2: no correlation coefficient is mentioned so it is difficult to conclude that it is not relevant. here again the spearman coefficient is needed
7. Ref. 25 (Maisonneuve et al. Cell 2013) has been retracted; avoid citing

Reviewer #1 (Remarks to the Author):

This paper explores cell-to-cell variability of *sasA*, which encodes a ppGpp synthase in *B. subtilis*. The authors show that *sasA* transcription in single cells has a very broad distribution across a population of cells and results from high extrinsic noise. They further show that the noise is primarily due to second-site phosphorylation at T101 in the response regulator WalR, with T101 phosphorylation controlled by the kinase PrpK and the phosphatase PrpC. Finally, they provide evidence that the noise has important physiological consequences, namely increased antibiotic tolerance. Over all I found the paper to be well written, the arguments compelling, and the results very interesting. The work provides a striking example of drug tolerance emerging from infrequent transcriptional events, a subject that is still in its nascent stages in both prokaryotic and eukaryotic biology. In addition, the regulation of cell-to-cell variability by second site phosphorylation of a response regulator seems quite novel.

I have only a few comments.

1) Fig. 2A. Why does the arrow that comes off of WalR split in two with one arrow pointing to PrkC? The figure makes it look like PrkC plays a role in dephosphorylating WalR, but there is no mention of this in the text. Some explanation or clarification in the figure caption should be provided.

Thank you for pointing out this point of confusion. It was not our intention to suggest that PrkC can dephosphorylate WalR. To increase clarity of the figure, **we have deleted the arrow cycling back to PrkC.**

2) Fig. 2B The histograms showing the distribution of autofluorescence should be shown (perhaps put it in the supplement). I realize that CDF for the autofluorescence is shown in Fig. S5 but it would be helpful to see the actual data binned as in Fig. 2B so that the reader can get a rough sense of the contribution of autofluorescence to histograms. Was the autofluorescence measured for each of the four strains shown in Fig. 2B or was a single strain used and the autofluorescence was assumed to be the same for the other strains?

Thank you for this suggestion. **We have now included a histogram of the autofluorescence in the supplement in figure S5.** In a separate experiment, the autofluorescence of each genotype was also confirmed to be similar to WT, and therefore, in subsequent experiments only a single non-fluorescent control was used. **This statement has now been added to the materials and methods (lines 59-61).**

Unfortunately, we no longer have the non-fluorescent data for all genotypes taken on the original flow cytometer which was used to generate Figure 2B (and the machine no longer exists). To show that the autofluorescence is similar among different strains, we have now taken data on a different flow cytometer. **This data is shown below, but not included in the revised manuscript to avoid confusion.**

Measurement of cellular autofluorescence in the YFP channel for WT (strain #JDB1772), $\Delta prpC$ (strain #JDB1773), and $\Delta(prpC-prkC)$ (strain #JDB1775) cultures. A biological replicate is also shown for WT for comparison.

3) Lines 174-175 I don't understand how the Komogorov-Smirnov test was used to compute p-values for the data in Fig. 2C. I understand how this test is used to compare distributions, but in this case, the data consists of a mean frequency of outliers based on a fixed threshold. Based on the data, the claim that PrkC has a significant effect on the mean frequency of outliers looks reasonable to me, but I suggest adding some sentences to the Materials and Methods explaining how the p-value was computed using the K-S test.

Thank you for this suggestion. We have now added a statement to materials and methods (SI lines # 137-144) clarifying how the KS testing was used to compare the observed distributions of % outliers between genotypes:

“Note on Significance testing for % outlier data shown in Figure 2

In Figure 2C at least 4 independent experiments measuring *sasA* expression were performed in WT, $\Delta prpC$, and $\Delta(prpC-prkC)$ in otherwise *walR*^{WT} and *walR*^{T101A} backgrounds. For each genotype, the % of cells above a fixed fluorescence threshold was measured and is shown as black dots. To test if the % values obtained for each genotype are statistically significant, we treated the independent measurements of %s as members of a distribution and performed Komogorov-Smirnov testing to determine the likelihood the % outliers observed for a comparison genotype pair could be drawn from the same distribution.”

4)To "deconvolve" the autofluorescence from the signal, why didn't the authors do an actual deconvolution using the measured autofluorescence distribution, instead of only removing the mean

and standard deviation components?

On our data set, brute force deconvolution in absence of any prior knowledge of the signal statistics is quite noisy and error prone. However, in bacteria, the statistics of protein copy numbers is known to be well described by the Gamma distribution, which is a great advantage as this distribution is fully specified in terms of its mean and variance. Given the measured mean and variance of the raw fluorescence intensity and of the autofluorescence, the mean and variance of the signal can be estimated with high accuracy. A Gamma estimate for the full statistics of the signal then follows immediately (see: "Mathematical details of deconvolution procedure" for more information), and we reported this estimate as the clean, deconvolved, signal. To further test that this estimate is internally consistent with our data, we reconvolved it with the autofluorescence and compared the statistics of the reconvolved sample with that of the measured raw fluorescence intensity as described below.

A note summarizing this has now been added to the SI.

5) I do not understand how the autofluorescence-free distributions were "re-convolved" with the autofluorescence data to generate Fig. S6. Was the autofluorescence-free distribution convolved with the actual measured distribution of autofluorescence and then smoothed? If not, where did the distribution for the autofluorescence data come from to take the convolution? Some more details on this should be included.

Indeed, to test its consistency, the estimated autofluorescence-free distribution was convolved with the actual measured distribution of autofluorescence. This was done by drawing two random fluorescence intensities: one from the estimated Gamma distribution for the clean signal as described above and one from the measured autofluorescence distribution. Adding these two random contributions gave a single, re-convolved, data point. Repeating this procedure for a number of times that equals the size of the

measured data set yielded the full re-convolved data set. The empirical distributions of the measured and re-convolved data sets were then compared (Fig. S7). No smoothing was applied.

The above explanation was indeed missing from the methods section: “Autofluorescence deconvolution method and validation”, but we have now added it.

Reviewer #2 (Remarks to the Author):

The manuscript presents observations of the high level of extrinsic noise in the expression of *sasA*, a ppGpp synthetase and the analysis of its origin and consequences. The authors dissect the role played by other factors known to affect *sasA*. They first measure the levels of SigM and conclude that it does not account for the variability. Then they analyze the role of the phosphorylation of WalR by the two component system PrpC/PrpK. They show that the deletion of *prpC* reduces the number of highly expressing cells. In order to compare not only the outliers but the whole distribution, they deconvolve each distribution with the strain’s auto-fluorescence signal. They expose the wt and *sas* deletion strain to an antibiotic and show a difference in the ratios of survival of high and low expressing *yfp* which suggest that the highly expressing *sasA* cells may be more tolerant than the low expressing *sasA* cells. Whereas the role of phosphorylation for generating variability is an interesting idea, here the analysis of mutants merely shows that the few outliers are missing but does not explain the interesting observation of the very large CV of *sasA*. The impact of such variability on persistence is potentially very interesting. However, there is no direct demonstration of the role of *sasA* or its phosphorylation on tolerance as detailed in the comments below.

1. The effect of the phosphorylation mutants seems to be measurable only for a very small portion of the population. What are the CVs of the distributions for the mutants?

The CVs of the mutants vary by about 30% as measured by flow cytometry. For example, we measured CV ~ 2.86 for WT, and ~ 2.2 for $\Delta prpC$. **This information is now included in the revised manuscript on lines #209-214.** (Also note that CVs measured by flow cytometry cannot be directly compared to CVs measured by microscopy, due to differences in sensitivity between the instruments.)

We note that the value of the CV is governed by the first two moments of the distribution, the mean and the variance. Since the mean expression of P_{sasA} is relatively low during log phase growth, the portion of the distribution that is the most physiologically interesting is the outliers (as shown in Fig. 4). **We therefore concentrated our analysis on factors that regulate the appearance of these outliers and their associated *sasA*-dependent physiology. We have reworked the first several paragraphs of the results section to make this focus clearer.**

Based on this question and the reviewer's summary comments above, we also extensively reworked the beginning of the results section to make it clear that we do not believe that PrkC activity fully accounts for the high CV of P_{sasA} . Indeed, that remains a bit of a mystery and beyond the scope of this work, as there appears to be several important contributing factors.

2. The survival of the wt and delta *sasA* are not mentioned, only the ratios of high to low, which seem arbitrary. How do the ratios change if the thresholds are changed?

Thank you for pointing out that this very important point was not clear. The bulk culture survivals were in the **original manuscript on line #224 (revised manuscript line #234).**

The thresholds in the original manuscript were chosen because the "high" expression gate corresponds to approximately 1% of the starting population and can be sorted in ~ 10 min. Higher thresholds than this result in very long sort times due to decreases in overall sorting efficiency, and also appeared to introduce unwanted variability in cell physiology. **A statement summarizing this has been added to the materials and methods (SI, line #45-48).**

To find the thresholds necessary to see a robust effect, we also performed several experiments at various lower cutoff thresholds for the "high" expression gate. Consistent with our model relaxing the high threshold to $\sim 60\%$ of the value in Fig. 4D, shows a much smaller survival advantage for WT cells. **This results in a reduction of the effect to $\sim 2.3 \pm 0.7$ fold for WT, and 0.9 ± 0.4 fold for $\Delta sasA$ (mean \pm range, 2 experiments), compared to a model prediction of 3.7 and 2.0-fold, respectively.** We concluded that experimentally measured survival advantage at this threshold was not strong enough to report as a robust effect. **Results of those experiments are now included in Fig. S13 and summarized in lines #266-269.**

Even though the results of these experiments produced a weak effect at best, they motivated us to pursue the project further and test the higher sorting thresholds reported in Fig. 4.

3. Importantly, in order to confer tolerance, the survival itself needs to be mentioned, not only the ratios. Is the *sasA* deletion mutant less tolerant? Are any of the other mutants tested more or less tolerant than the wt? This information is crucial in order to understand the role of *sasA* noise in tolerance as well as the role played by phosphorylation.

As above, the bulk culture survivals were in the **original manuscript on line #224 (revised manuscript line #234)**. To highlight that in **bulk** culture we do not observe a robust difference in survival between WT and $\Delta sasA$ or $\Delta(prpC-prkC)$ we have now included **Fig. S11A which includes the survival assayed by CFUs**.

Note that consistent with our results in Fig. 4 which show that the survival depends on pre-treatment *sasA* expression levels, we do not observe a difference in tolerance at the population level, including for the mutants. However, we do not expect that bulk population CFU assays would be sensitive enough to pick up differences consistent with our results in Fig. 4 where the *sasA*-dependent effect is most apparent for the tails of the distribution. (We expect, for example, the upper 1% of the population has a ~10-fold increase in survival.)

To address the reviewer's question at the single-cell-level, we performed a similar experiment to Fig. 4AB, in the $\Delta(prpC-prkC)$ background. Consistent with Figs. 2-3, we found that there are more cells with high levels of *sasA* expression in the $\Delta(prpC-prkC)$ background prior to ciprofloxacin treatment. We also found that ciprofloxacin treatment resulted in a strong enrichment of bright cells within the population. However, as in the case of WT, we found that this effect was only partially *sasA*-mediated. This is perhaps unsurprising given the known tendency for mutants to be pleiotropic and the diverse list of putative targets of PrkC. **This suggests that there is a difference in tolerance of the $\Delta(prpC-prkC)$ mutant observable at the single-cell-level, but that as in the case of WT populations, the effect of PrkC on tolerance is only partially *sasA*-mediated. We have included this data in Fig. S12 and a summary of these results in lines 244-249 in the revised manuscript.**

4. The authors conclude that the increased *sasA* expression in the FACs is due to the enrichment of pre-induced cells. However, the FACS does not enrich for surviving bacteria, does it? Ciprofloxacin killing typically does not result in cell lysis so that both dead and viable cells will be measured by FACS.

Thank you for raising this important point. Under our experimental conditions we observed significant cell lysis for ciprofloxacin killing: 1) we spun down treated cultures and observed a significant amount of lysed cells and debris 2) We also observe a drop in OD₆₀₀ correlated with a drop in CFUs. We have now added supplemental figure S11 and associated reference to the figure at line #234 in the main text.

(For the purposes of our response, we assumed that the reviewer is referring to the flow cytometry experiments in Fig 4A&B, not the FACS experiments in Fig.4D&E. The important distinction is that the FACS experiment in Fig. 4 D&E is only performed on cells prior to ciprofloxacin treatment, followed by plating for CFUs. This experiment was specifically designed not be biased by any potential issues involving measurements on treated cells. Indeed, we view panels A&B as suggestive, but D&E as the key confirming experiments for that reason.)

5. Fig. S1: it is not clear why the data is presented on a linear vs log scale, and not in log-log scales as in Fig. 1. Also, the correlation coefficients should be calculated over the whole range and compared to the correlation of the two *sas* promoters (0.9 Pearson). BTW, because of the outliers, it may be better to calculate the Spearman coefficients.

Thank you for pointing out this point of confusion. Originally, we plotted on a linear scale simply to increase the visibility of the points as the P_{veg} data spans ~half a decade, and P_{sasA} spans 5 decades. However, as the reviewer noted, this may cause confusion, so **we have changed the plot to a log-log scale**. (To increase the visibility of the individual data points on a log-log plot, we have also decreased the marker size of the individual data points.)

We used the pearson correlation coefficient to compare the *sasA* promoters because we were interested in determining the strength of the linear relationship between them. Since the test for extrinsic noise relies on testing whether the two promoters largely behave the same, the linear test is appropriate. (This is also standard in the field.) The spearman correlation coefficient tests for something slightly different: it provides a measure for the monotonic relationship between the two promoters. This is likely more appropriate and interesting when comparing between different promoters (*e.g. sasA* and *sigM*.) As the reviewer points out, we cannot directly compare values for spearman and pearson correlation coefficients. It is also true that the outliers may be influencing the pearson correlation coefficients. Since both pearson and spearman correlation coefficients provide different types of information, and pearson is the standard in the field for examining sources of noise, **we have elected to provide both the pearson and spearman coefficients throughout the paper**, allowing the reader to make the appropriate comparison of interest.

6. Fig. S2: no correlation coefficient is mentioned so it is difficult to conclude that it is not relevant. here again the spearman coefficient is needed

The correlation coefficient for the data in Fig. S2 was on lines 161-162 of the supplemental information in the original manuscript (lines #188-189 in the revised SI). As explained above, we have also added the spearman coefficient.

7. Ref. 25 (Maisonneuve et al. Cell 2013) has been retracted; avoid citing
Thank you for pointing out this oversight. (The retraction occurred since the paper was originally drafted.) **We have removed the reference and the associated review (#s 21 and 25 in the original manuscript).**

Reviewers' comments:

Reviewer #1 (Remarks to the Author):

I am happy with the authors' responses to reviewer comments and the revised manuscript and have no additional concerns.

Reviewer #2 (Remarks to the Author):

The authors have significantly improved their manuscript, more clearly stating that their analysis cannot explain the full range of the *sasA* variability, but shows that a small part of the variability can be attributed to the phosphorylation of WalR. More specifically, they focus on characterizing the effect of the phosphorylation on the outliers of the distribution which constitute about 0.3% of the population. By using the Pveg-mcherry control, they show that the overexpression of fluorescence is specific to the *sasA* promoter and can be increased or decreased by specific deletions. This part is convincing. A minor comment is that since the percentage of the outliers is low, the authors should better describe how the cultures were grown prior to the fluorescence measurements, and show that the outliers are not from the carryover of previous growth conditions.

My main concern remains the analysis of the persistence assays: I apologize for missing the line stating that the survival is the same with or without *sasA*, I mistakenly was looking for data showing the opposite. Given that the *sasA* deletion has the same survival as the wt, how can the authors write (line 242) "...demonstrating that *sasA* has a significant contribution to survival after ciprofloxacin treatment"? or (line 326) "*sasA* exerts a continuous effect proportional to its level on physiological processes that effect ciprofloxacin killing"? The simple conclusion from the equal survival with and without *sasA* is that *sasA* has no effect on survival. The fact that the fluorescence distributions have a mild change after the antibiotic treatment may be related to an indirect effect for example whether the dead cells are fully lysed or not, which may depend on the genetic background, but this has no direct link to persistence under antibiotic treatments. Furthermore, it may be that any fluorescent marker (such as the Pveg-mcherry control) would show the same effect. Have the authors tried to keep the antibiotic treatment for longer times, where the outliers' contribution to survival may be directly observed? Without a direct effect of *sasA* deletion on the survival level, the link to persistence is still missing.

Reviewer #2: My main concern remains the analysis of the persistence assays: I apologize for missing the line stating that the survival is the same with or without *sasA*, I mistakenly was looking for data showing the opposite. Given that the *sasA* deletion has the same survival as the wt, how can the authors write (line 242) "...demonstrating that *sasA* has a significant contribution to survival after ciprofloxacin treatment"? or (line 326) "*sasA* exerts a continuous effect proportional to its level on physiological processes that effect ciprofloxacin killing"? The simple conclusion from the equal survival with and without *sasA* is that *sasA* has no effect on survival. The fact that the fluorescence distributions have a mild change after the antibiotic treatment may be related to an indirect effect for example whether the dead cells are fully lysed or not, which may depend on the genetic background, but this has no direct link to persistence under antibiotic treatments. Furthermore, it may be that any fluorescent marker (such as the Pveg-mcherry control) would show the same effect. Have the authors tried to keep the antibiotic treatment for longer times, where the outliers' contribution to survival may be directly observed? Without a direct effect of *sasA* deletion on the survival level, the link to persistence is still missing.

As noted above, we are not studying persistence. Persistence has a well-defined quantitative definition and we do not claim to be studying it. The term is not mentioned in the manuscript. We further note that the data in Fig. S11 illustrates that tolerance (the term we use) is a better description of our observed phenomena.

The reviewer's comments suggest that he/she has missed the high-level point that there is a difference between measurements of bulk culture survival and measurements of survival at the single cell level and that we present both in the paper. **We do observe differences in survival at the single cell level that are proportional to *sasA* expression (Fig. 4). Bulk culture survival measurements are not sensitive enough to observe this**, a point that we reiterated several times in our last response.

Furthermore, we do not claim that WT and $\Delta sasA$ backgrounds have the same survival: we write on line 234 that when measured in bulk culture WT and $\Delta sasA$ have approximately the same survival "~99% killing in both WT and $\Delta sasA$ backgrounds (Fig. S11)." The legend in Figure S11A states "ciprofloxacin treatment results in ~1-2% survival of WT, $\Delta sasA$ populations as assayed by colony forming units."

Our previous response attempted to also clarify this to the reviewer by highlighting the word "bulk" and repeating it throughout our last response (replies to points #2 and 3). We also already included an entire paragraph (reply to point #3, second paragraph) detailing that bulk CFU measurements are not sensitive enough to measure the effect of *sasA* expression on tolerance, so we used FACS to sort cells by expression and then perform CFU based killing assays. We also noted (and note again now) that we sorted WT and $\Delta sasA$ populations prior to antibiotic treatment and then plated for CFUs – an assay that is **not confounded by lysis or changes in fluorescence distributions as the fluorescence is only measured prior to antibiotic treatment**. In those experiments we confirmed that **there is a robust *sasA*-dependent effect on survival (Fig. 4E)**.

To illustrate why the bulk culture and FACS experiments are in agreement with a *sasA*-expression-effect on survival, consider the **following simple calculation comparing bulk culture survival with experimental error**:

For the purposes of this calculation, we will use the experimentally observed values reported in the paper: we observe ~1-2% survival for the bulk population, a similar number for cells with “low fluorescence” (as expected, since they make up the majority of the population), and approximately a 5 fold increase (~10% survival) for the “high fluorescence” cells in the FACS experiment. (This will likely overestimate/provide an upper bound for the max expected difference observable in bulk culture survival between WT and $\Delta sasA$ backgrounds.)

If we were to estimate that ~99% of a culture of 10^6 bacteria has a 2% survival, roughly ~ 19,800 cells survive ($0.99 * 10^6 * 0.02 = 19800$). The top 1% of that culture would have an increased survival (in WT) with 1000 cells surviving ($0.01 * 10^6 * 0.1 = 1000$). Bulk culture survival is assayed by CFUs, where we observe experimental error of approximately ~0.1-0.5% in bulk culture survival (Fig. S11), corresponding to inter-experiment variability of +/-1000-5000 cells ($0.001 * 10^6 = 1000$; $0.005 * 10^6 = 5000$). Therefore, the observed variability between experiments is 2-10x the contribution of the *sasA* expression outliers in bulk culture.

Therefore, to further emphasize this in the manuscript, we propose adding the following:

Line 234: “when measured at the bulk population level”

Line 258: “Importantly, sorting cells by *sasA* expression level prior to ciprofloxacin treatment allows more sensitive measurements of the survival advantage conferred by SasA than bulk population level CFU measurements are capable of. Additionally, cells were measured and sorted prior to antibiotic treatment and assayed by CFUs to avoid confounding survival measurements with changes in cell physiology, fluorescence protein expression, and lysis in response to ciprofloxacin.”

Line 260: We observed cells with high levels of *sasA* expression had a survival advantage of 9.5 ± 0.6 for WT, but only 1.8 ± 0.7 in $\Delta sasA$, demonstrating that cells with high levels of *sasA* expression prior to antibiotic treatment preferentially survive.

Reviewer #2: A minor comment is that since the percentage of the outliers is low, the authors should better describe how the cultures were grown prior to the fluorescence measurements, and show that the outliers are not from the carryover of previous growth conditions.

In response we propose adding the following statement to the materials and methods:

SI line 10: For every experiment, strains were streaked out from frozen stocks on LB Lennox and grown overnight (~15hrs) at 37C. Single colonies were used to inoculate 3 ml liquid cultures in S7 (supplemented with inducers as indicated) and cultures were grown to early log phase.

We note that the hypothesis that the outliers are carryover from previous growth conditions is not supported by the data already in the paper: for example, the demonstration that their frequency can be

strongly reduced by expression of PrkC *in trans* (Fig. S6). Experiments to definitively rule out a hypothesis for their origin will likely be quite challenging and is beyond the scope of this work.

REVIEWERS' COMMENTS:

Reviewer #2 (Remarks to the Author):

The authors have clarified an important point that was buried before and rephrased the strong sentences that present *sasA* as an important factor for tolerance. As they fail to detect any effect of *sasA* deletion on the survival of the bulk population, they restrict their analysis only on the survival of the extremity of the fluorescence expression distribution and compare the survival of cells with mean expression to that of very high expression. They show that the highly expressing *sasA* cells in the tail of the distribution are more tolerant than the ones with mean expression. However, since the main results of the work rely on the tail of the distribution representing less than 1% of the population, it is surprising that the authors chose to work on cultures initiated from a colony in a small volume (3ml). Such cultures have been shown to be inherently heterogeneous because of carry over effect. This effect may depend on conditions/strains so that the observation that it is different in the *PrkC* in trans does not rule it out. The best would be that the authors repeat the survival of the high expressing *sasA* tail from a similar culture but diluting it 1:1000 after reaching OD 0.1, and regrowing it again to 0.1, making sure that the tail that they evaluate does not originate from bacteria still in the stationary state of the colony, where *sasA* expression may have been triggered. Alternatively, the authors should mention in the discussion that they cannot rule out that the highly expressing cells originate from the colony itself.

Response to referee comments:

Reviewer #2 (Remarks to the Author):

The authors have clarified an important point that was buried before and rephrased the strong sentences that present *sasA* as an important factor for tolerance. As they fail to detect any effect of *sasA* deletion on the survival of the bulk population, they restrict their analysis only on the survival of the extremity of the fluorescence expression distribution and compare the survival of cells with mean expression to that of very high expression. They show that the highly expressing *sasA* cells in the tail of the distribution are more tolerant than the ones with mean expression. However, since the main results of the work rely on the tail of the distribution representing less than 1% of the population, it is surprising that the authors chose to work on cultures initiated from a colony in a small volume (3ml). Such cultures have been shown to be inherently heterogeneous because of carry over effect. This effect may depend on conditions/strains so that the observation that it is different in the *PrkC* in trans does not rule it out. The best would be that the authors repeat the survival of the high expressing *sasA* tail from a similar culture but diluting it 1:1000 after reaching OD 0.1, and regrowing it again to 0.1, making sure that the tail that they evaluate does not originate from bacteria still in the stationary state of the colony, where *sasA* expression may have been triggered. Alternatively, the authors should mention in the discussion that they cannot rule out that the highly expressing cells originate from the colony itself.

Unfortunately, we cannot repeat the FACS experiment to measure survival, as the exact machine used to perform the experiments in the paper no longer exists. Therefore, we have added the following statement to the discussion:

“This demonstrates that *PrkC* activity is likely heterogeneous in our culture condition, but does not pinpoint the cause or origin of the variability. For example, from this study we cannot rule out that cells with high levels of *sasA* expression originate in a *PrkC*-dependent manner from the colony itself.”